# Quantum advantage for learning shallow neural networks with natural data distributions

Laura Lewis [1,2,3,4] ✉, Dar Gilboa[1] & Jarrod R. McClean [1]

Without large quantum computers to empirically evaluate performance, theoretical frameworks such as the quantum statistical query (QSQ) are a primary tool to study quantum algorithms for learning classical functions and search for quantum advantage in machine learning tasks. However, we only understand quantum advantage in this model at two extremes: either exponential advantages for uniform input distributions or no advantage for arbitrary distributions. Our work helps close the gap between these two regimes by designing an efficient quantum algorithm for learning periodic neurons in the QSQ model over a variety of non-uniform distributions and the first explicit treatment of real-valued functions. We prove that this problem is hard not only for classical gradient-based algorithms, which are the workhorses of machine learning, but also for a more general class of SQ algorithms, establishing an exponential quantum advantage.

Machine learning (ML) is currently experiencing explosive success, made possible by an overwhelming growth of compute power, data availability, and improved models[1–4]. In parallel, quantum technology is also witnessing remarkable progress, including breakthroughs in quantum error correction[5–19] and demonstrations of computations beyond the known limits of classical computers[6,20–26]. Given that our universe is inherently quantum, it is natural to consider leveraging powerful quantum computers for ML tasks, in hopes of new scientific advancements[24,27–41]. However, modern classical ML is mainly driven by empirical success, extending far beyond our theoretical understanding. In contrast, quantum technologies are still in their infancy, where we cannot yet accurately train and test large quantum ML models. Thus, we must rely on the rigorous frameworks of learning theory to characterize the performance of quantum learning algorithms and their potential advantage over classical learners.

One possible avenue for quantum advantage is to use quantum algorithms to learn classical objects, e.g., classical functions[42–57] or distributions[58–60]. Such results commonly consider the quantum counterparts of frameworks, such as probably approximately correct (PAC)[61] and statistical query (SQ) learning[62], appropriately called quantum PAC[42] and quantum SQ (QSQ)[44], respectively. In particular,

some exciting results show that there exist function classes for which quantum PAC/QSQ algorithms can provide exponential sample complexity advantages over classical learners when the input data distribution is uniform[42,44–49,52,53]. This is in stark contrast to the seminal result proving there is no quantum advantage for arbitrary distributions[55,63–65]. The void between exponential advantages on idealized uniform distributions and no advantage on potentially adversarial distributions leaves a large gap in our understanding of quantum learning advantages. These results also highlight the challenges in analyzing quantum advantage for empirical data distributions and mirror results in classical ML, where there exists problems that are NP-complete for arbitrary distributions but easy for the distribution-specific case[66–69]. Moreover, to our knowledge, all results in quantum learning theory to date focus on Boolean or discrete functions, while the majority of large-scale ML focuses on real-valued functions. Together, these two points raise our central question:

Are there classes of real-valued functions and non-uniform distributions for which quantum data is advantageous?

These are also stated as two open questions in ref. 70. Here, by quantum data, we mean classical functions over distributions encoded into so-called quantum example states, as in quantum PAC and QSQ

[1]Google Quantum AI, Venice, CA, USA. [2]University of Cambridge, Cambridge, UK. [3]University of Edinburgh, Edinburgh, UK. [4]Present address: University of California, Berkeley, Berkeley, USA. ✉e-mail: llewis@alumni.caltech.edu

learning. We provide a new perspective on when these states might arise naturally later in the work.

While some results consider learning Boolean functions over $c$-bounded product distributions[50,51], proving quantum advantages for more general non-uniform distributions still remains open. Moreover, for other forms of quantum data, such as expectation values of ground states, quantum advantages for learning over non-uniform distributions have been explored[71]. However, this is incomparable to the present work, where we focus on classical functions encoded in quantum example states.

In this work, we provide a positive answer to our central question by efficiently learning real-valued functions that are a composition of a periodic function and a linear function in the QSQ model over a broad range of non-uniform distributions, which includes Gaussian, generalized Gaussian[72], and logistic distributions. These distributions are practically relevant with generalized Gaussian and logistic distributions finding applications in, e.g., image processing[73–75] and population growth[76–79], respectively. Moreover, note that success in the QSQ model automatically implies success in the quantum PAC model, as the QSQ model is strictly weaker because it does not allow entangled measurements[80].

We highlight that the function class we consider is well-studied in the classical ML literature[81–84]. There, such functions – called *cosine neurons* or, more generally, *periodic neurons* – are commonly analyzed, as they form the basic structure of neural networks with periodic activation functions[85–90] and can be seen as an extension of generalized linear models[91,92]. In particular, ref. 82 proves that any gradient-based classical algorithm cannot learn periodic neurons when the input data distribution has a sufficiently sparse Fourier transform, which is satisfied by many natural distributions, e.g., Gaussians, mixtures of Gaussians, Schwartz functions[93], etc. We strengthen their proof to apply to our specific parameter choices that focus on the regime of quantum advantage. Furthermore, although gradient methods are perhaps the most popular in classical ML, there is strong evidence for classical hardness beyond gradient methods. In fact, we extend the classical hardness to hold for a more general class of algorithms performing correlational SQs[94,95]. Additionally, ref. 83 shows an exponential lower bound for any classical SQ algorithm learning this function class with respect to any log-concave distribution. Ref. 84 extends the hardness to any polynomial time classical algorithm learning under small amounts of noise and over Gaussian distributions, assuming the hardness of solving worst-case lattice problems[96,97]. These results[83,84] do not directly apply to our setting due to a difference between the parameter regimes needed for quantum advantage versus classical hardness, but we expect classical hardness to still hold in this regime and leave this generalization open to future work.

Our algorithm uses a polynomial number of QSQs and iterations of gradient descent, resulting in a quantum advantage over any classical gradient-based algorithm for sufficiently Fourier-sparse input distributions[82]. Here, the classical algorithms considered are any algorithms that utilize approximate gradients of an average loss function, which includes, e.g., neural networks. Concretely, we obtain an exponential quantum advantage against classical gradient methods for Gaussian, generalized Gaussian, and logistic distributions. For Gaussian distributions, we additionally strengthen classical hardness to hold against a natural restriction of SQ algorithms (namely, correlational SQ algorithms[94,95]), which includes gradient methods, dimension reduction, and moment-based methods. To our knowledge, this is the first result in quantum learning of classical functions that explicitly considers real-valued functions. Figure 1 illustrates a schematic overview of our work.

## Results

In this section, we introduce the task of learning periodic neurons and show that it is classically hard for a broad class of powerful algorithms.

Then, we detail our quantum algorithm that solves this problem efficiently, exhibiting an exponential quantum advantage.

## Problem definition

In this section, we define common access models in (quantum) learning theory and describe our learning problem more formally. We refer to Supplementary Note 1A and Supplementary Note 2 for further details.

We aim to learn a collection of functions $\mathcal{C} \subseteq \{c : \mathcal{X} \to \mathcal{Y}\}$ called the *concept class*, where $\mathcal{X}, \mathcal{Y}$ are the input/output spaces, respectively. In particular, given some form of access to an unknown concept $c^\star \in \mathcal{C}$, we want to learn an approximation of $c^\star$ with high probability. Typically in learning theory, one considers Boolean functions with $\mathcal{X} = \{0,1\}^d, \mathcal{Y} = \{0,1\}$. Importantly, in this work, we instead consider $\mathcal{X} = \mathbb{R}^d, \mathcal{Y} = \mathbb{R}$.

In the classical PAC model[61], the learning algorithm is given labeled random examples $(x_i, c^\star(x_i))_{i=1}^N$, where the $x_i$ are sampled from a distribution $\mathcal{D}$ over $\mathcal{X}$ and $c^\star \in \mathcal{C}$ is an unknown target function. The SQ model[62] is weaker than PAC, where, instead of direct access to the examples, the learning algorithm can only obtain noisy expectation values of functions of the data. This was originally proposed to model learning given noisy examples, and commonly used algorithms, such as stochastic gradient descent[98], Markov chain Monte Carlo methods[99,100], and simulated annealing[101,102] can be implemented in this model.

We also consider the *correlational* SQ model[94,95]. This is a restriction of general SQs in which queries are only allowed to act on the input space $\mathcal{X}$, not the labeled output space. We define this more precisely in Supplementary Note 1A. Correlational SQs include gradient methods, dimension reduction, and moment-based methods as special cases. In the case of Boolean functions, correlational SQs and general SQs are in fact equivalent[94], but there exist separations between them for real functions[103–105].

The quantum PAC and QSQ models are natural generalizations of these settings. In the quantum PAC model[42], the learning algorithm is given copies of the quantum example state

$$|c^\star\rangle \triangleq \sum_{x \in \mathcal{X}} \sqrt{\mathcal{D}(x)} |x\rangle |c^\star(x)\rangle, \tag{1}$$

where $\mathcal{D}$ is again some probability distribution. We note that there are some restrictions on the distributions $\mathcal{D}$ for which one can efficiently prepare this state and discuss this later. Also notice that upon measuring a quantum example state, one obtains $(x, c^\star(x))$ for $x$ sampled from the distribution $\mathcal{D}$, hence recovering the classical PAC examples. For QSQ access[44], the learner queries an observable $O$ and receives an approximation of the expectation value $\langle c^\star | O | c^\star \rangle$. This is weaker than the quantum PAC model due to the inability to take entangled measurements across multiple copies of $|c^\star\rangle$[80]. Notice also that because $\mathcal{X}$ is a continuous space in our setting, these definitions require discretization/truncation, which we discuss further in the Methods and Supplementary Note 1 A. In all aforementioned cases, the goal is to learn the unknown function $c^\star$ approximately with high probability using as few examples/queries as possible.

We are interested in learning a concept class consisting of functions that are a composition of a periodic function and a linear function. In other words, these are functions that can be represented as a single-layer neural network with a periodic activation function, hence dubbed periodic neurons. This ansatz is quite powerful and in some cases is able to achieve universal function approximation[106–108]. Moreover, the periodic neuron has known relationships to important complexity theoretic problems[84,109].

Explicitly, let $d \geq 1$ denote the input dimension and let $\mathbb{S}^{d-1}$ denote the $(d-1)$-dimensional unit sphere. Then, our concept class is defined

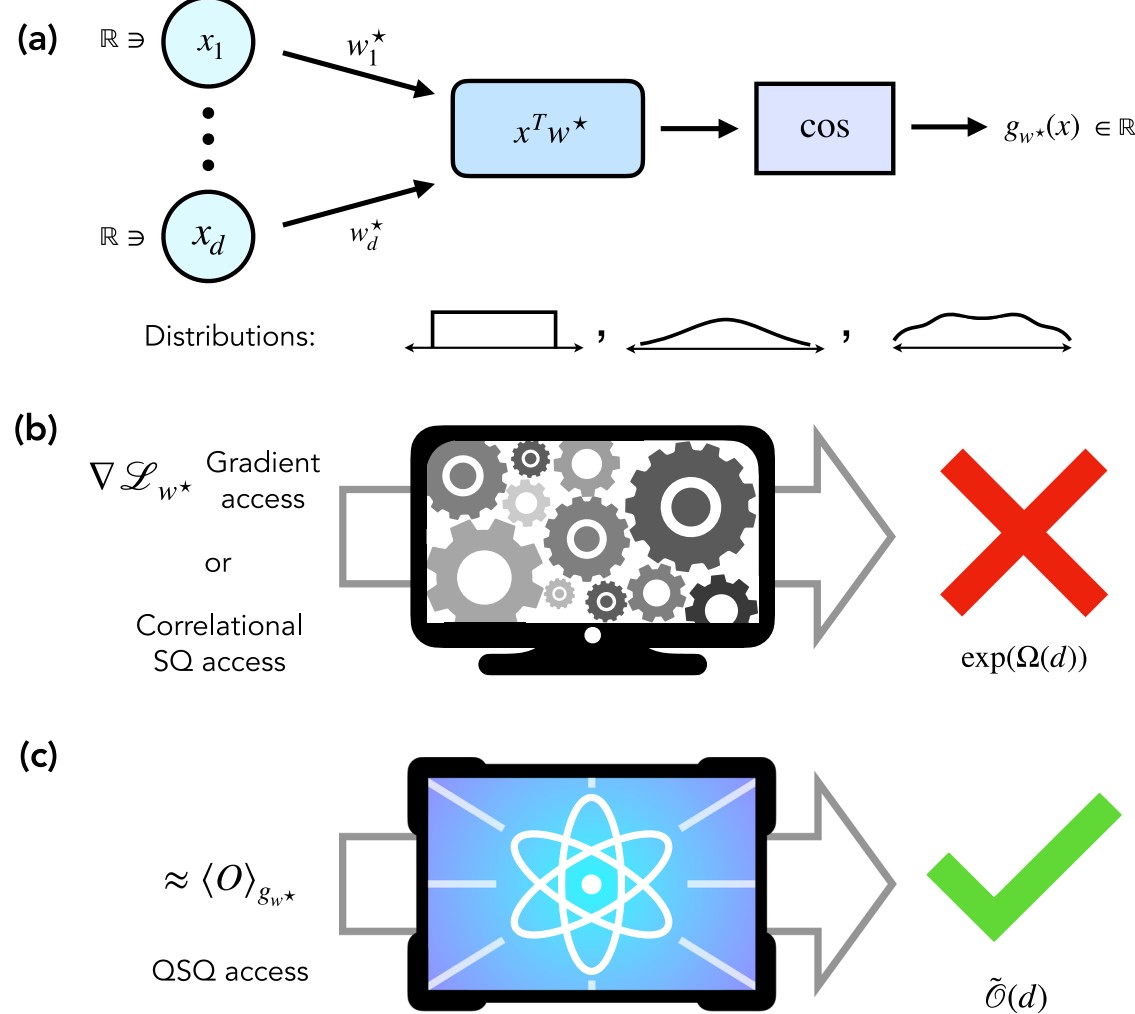

**Fig. 1 | Overview of results. a** Target function and input distributions. Given an input vector $x \in \mathbb{R}^d$, we consider learning functions of the form $g_{w^\star}(x) = \cos(x^\top w^\star)$, where $w^\star \in \mathbb{R}^d$ is an unknown vector. Our illustration emphasizes their connection with classical deep learning, where they are called cosine neurons. We also consider more general periodic neurons, which one can view as linear combinations of cosine neurons with unknown weights. We consider input distributions, such as uniform, Gaussians, and more general distributions which are sufficiently flat, as characterized by technical conditions specified in Supplementary Note 5. **b** Classical hardness. We strengthen the arguments of[82] to show that classical gradient and correlational SQ methods require an exponential number of iterations (i.e., an exponential number of gradient samples) in the dimension of the problem and the norm $R_w$ of $w^\star$ to learn these functions. **c** Quantum algorithm. In contrast, our new quantum algorithm using QSQs is exponentially more efficient with respect to both time and sample complexity.

as

$$\mathcal{C} \triangleq \{g_{w^\star} : \mathbb{R}^d \to [-1, 1] : g_{w^\star}(x) = \tilde{g}(x^\top w^\star), w^\star \in R_w \mathbb{S}^{d-1}\}, \quad (2)$$

where $R_w > 0$ is the norm of the unknown vector $w^\star$ and $\tilde{g} : \mathbb{R} \to [-1, 1]$ is a periodic function of period 1 that can be written as

$$\tilde{g}(y) = \sum_{j=1}^{D} \beta_j^\star \cos(2\pi j y), \quad ||\beta^\star||_1 = 1 \quad (3)$$

for some constant $D > 0$ and unknown parameters $\beta_j^\star \in \mathbb{R}$. In other words, our target functions $g_{w^\star}$ are defined as follows. First, consider an unknown vector $w^\star$ of norm $R_w$, and consider the linear function $x^\top w^\star$ defined by this coefficient vector. Then, compose this linear function with a linear combination of cosines, where the weights $\beta_j^\star$ are unknown. In our analysis, we have additional constraints on the vector $w^\star$, e.g., restricted to the positive orthant and bounded away from 0, but for simplicity of presentation, we omit this detail in the main text. We direct the reader to Supplementary Note 2 for more details.

To learn a target concept $g_{w^\star}$ with respect to a distribution $\mathcal{D}$, we want to find a good predictor $f_\theta(x)$ which minimizes the objective function

$$\min_{\theta \in \Theta} \mathcal{L}_{w^\star}(\theta) \triangleq \min_{\theta \in \Theta} \mathbb{E}_{x \sim \mathcal{D}}[(f_\theta(x) - g_{w^\star}(x))^2], \quad (4)$$

where $\theta$ are some tunable parameters. Namely, for a given $\epsilon > 0$, we want to find parameters $\hat{\theta}$ such that $\mathcal{L}_{w^\star}(\hat{\theta}) \leq \epsilon$. Classically, we consider algorithms that have access to gradients of this loss function and can compute it for a given choice of parameters $\theta$. Our quantum algorithm additionally has QSQ access to the (discretized/truncated) example state $|g_{w^\star}\rangle$. Here, discretization is necessary to encode the continuous outputs of the target function into a discrete quantum state, and we similarly require truncation to ensure that the superposition is not over an infinite space.

While gradient access is more restrictive than general classical SQ algorithms, SQ algorithms include gradient methods as a special case[98]. Gradient-based algorithms are also the most widely used methods to train neural networks in practice. Moreover, for Gaussian distributions, we extend classical hardness to hold against

correlational SQ algorithms. These are more restrictive than general SQ algorithms[103–105], but we nevertheless view this as an important step towards proving SQ hardness. As discussed above, there is also strong evidence that the problem remains hard for general SQ algorithms and even all efficient classical algorithms[83,84]. In fact, the techniques for proving hardness against gradient methods[82] are similar to those for existing SQ hardness results[62,110].

## Classical hardness

Previous work from the classical literature[81,82] shows that learning periodic neurons as described in the previous section is hard for classical gradient methods, which includes powerful algorithms, such as classical neural networks. This result holds for any input distribution that is sufficiently sparse in Fourier space, defined by the notion of $\epsilon(r)$-Fourier-concentration. Intuitively, $\epsilon(r)$ is a function which characterizes how quickly the Fourier transform of the density function decays. We define Fourier concentration formally in Definition 3 in Supplementary Note 3. In Supplementary Note 3 A, we strengthen the proof from[82] to show that classical hardness still holds for our additional constraints on the vector $w^\star$.

**Theorem 1.** (A variant of Theorem 4 in ref. [82]; Informal) Let $g_{w^\star} : \mathbb{R}^d \to [-1, 1] \sim \text{Unif}(\mathcal{C})$ be a uniformly sampled target function, where the unknown vector $w^\star \in \mathbb{R}^d$ has norm $R_w$. Consider an input distribution whose density $\varphi^2$ can be written as a square of a function $\varphi$ and is $\epsilon(r)$-Fourier-concentrated. Let $\epsilon' = \sqrt[3]{c_1(\exp(-c_2 d) + \sum_{n=1}^{\infty}\epsilon(nR_w/4)}$ for constants $c_1$, $c_2$. Then, any classical gradient-based algorithm requires at least $p/\epsilon'$ gradient samples with $\epsilon'$ precision to learn $g_{w^\star}$ with probability $1 - p$ over the choice of $g_{w^\star}$.

Note that there are similar classical hardness results which hold for any 1-Lipschitz loss function[81], rather than the squared loss $\mathcal{L}_{w^\star}$ from (4). This theorem tells us that if the function $\epsilon(r)$ decays rapidly with $r$, then unless the number of gradient samples is extremely large or the noise in the problem is unrealistically small, a classical gradient-based algorithm cannot learn the concept class $\mathcal{C}$ from (2). We note that we only obtain a meaningful lower bound when $\epsilon(r)$ decays sufficiently quickly such that the infinite sum in the expression for $\epsilon'$ converges. This is guaranteed when the Fourier transform of the input distribution has sharply decreasing tails. For instance, for Gaussian distributions, the number of samples must scale as $\exp(\Omega(\min(d, R_w^2)))$. Here, the classical hardness stems from the gradient of the loss function concentrating around a fixed value, which, in turn, is due to the Fourier sparsity of the input distribution and target functions.

Furthermore, for the case of Gaussian distributions, we strengthen the classical hardness to hold against any classical algorithm which has access to correlational SQs[94,111]. This model is more general than gradient methods but is still a restriction of general SQs. We view our proof of classical hardness against such algorithms as an important step towards general SQ hardness.

Additionally, it is interesting to observe that only one type of query made by our quantum algorithm is not a correlational QSQ, i.e., observables of the form $O \otimes I$, where the identity acts on the output register. Thus, one might argue that considering only correlational SQs for classical hardness is not a significantly unfair comparison. We prove the following theorem.

**Theorem 2.** (Correlational SQ Hardness; Informal) Consider a Gaussian distribution with a sufficiently large variance. Then, any classical algorithm using correlational SQs to $\mathcal{C}$ with respect to this distribution requires at least $2^{\Omega(d)}$ queries to learn $\mathcal{C}$ to error $\epsilon$.

The full theorem is stated in Theorem 2 in Supplementary Note 3B. Importantly, we highlight that the condition on the variance of the Gaussian distribution is satisfied by our quantum algorithm

presented in the next section. Thus, classical hardness holds in the same regime as our efficient quantum algorithm. We also remark that, previously, classical learning theorists have shown similar correlational SQ lower bounds for learning single-layer neural networks[112,113]. However, these works consider different activation functions, so their results are not immediately applicable. We prove Theorem 2 by lower bounding the statistical dimension[95,110,114] of $\mathcal{C}$, which captures the difficulty of learning a concept class, similarly to the more commonly known VC dimension. The proof is provided in Supplementary Note 3B.

## Quantum algorithm

In contrast, the complexity of our quantum algorithm scales only polynomially in $d$ and polylogarithmically in $R_w$, since quantum algorithms can overcome and in fact leverage this Fourier sparsity via the quantum Fourier transform. We state our guarantee first for the uniform distribution. We highlight that while classical hardness for gradient methods holds on average over a uniform choice of $g_{w^\star}$ from the concept class, the guarantee for our quantum algorithm applies in the stronger worst-case setting, i.e., it holds for any fixed $g_{w^\star}$. Our correlational SQ hardness result holds in the worst-case setting as well.

**Theorem 3.** (Uniform distribution; Informal Version of Theorem 4 in Supplementary Note 4) Let $\epsilon > 0$, and let $\varphi^2$ be the uniform distribution. Let $g_{w^\star} : \mathbb{R}^d \to [-1, 1] \in \mathcal{C}$ be a target function for an unknown vector $w^\star \in \mathbb{R}^d$ with norm $R_w$. Then, there exists a quantum algorithm with QSQ access to a suitably discretized quantum example state $|g_{w^\star}\rangle$ that can efficiently find parameters $\hat{\beta} \in \mathbb{R}^D$ such that $\mathcal{L}_{w^\star}(\hat{\beta}) \le \epsilon$ with high probability using

$$N = \mathcal{O}(dD \, \text{polylog}(d, D, R_w, 1/\epsilon)) \tag{5}$$

QSQs and $t = \Theta(\log(D/\epsilon))$ iterations of gradient descent.

The detailed theorem statement is given in Theorem 4 in Supplementary Note 4. There, we also specify the QSQ noise tolerance needed explicitly. We highlight that it is only required to scale inverse polynomially in all parameters. We are also able to achieve a similar complexity for learning with respect to "sufficiently flat" non-uniform distributions. The precise technical conditions needed are stated in Supplementary Note 5. In particular, we show in Supplementary Note 5 that these conditions are satisfied for three practically-relevant classes of distributions: Gaussians, generalized Gaussians[72], and logistic distributions. The wide applicability of Gaussian distributions is clear, and generalized Gaussians and logistic distributions have applications in image processing[73–75] and population growth[76–79], respectively.

The flatness property we require is typically satisfied by taking a distribution's scale parameter large enough. Nevertheless, the condition still permits distributions that deviate significantly from uniform. For example, for Gaussian distributions, we need that the variance $\sigma$ is large enough such that $e^{-x^2/\sigma^2}$ is point-wise close to 1, e.g., $|e^{-x^2/\sigma^2} - 1| \le 1/10$ over our truncated space, and that the derivative of the density function is not too large. These conditions are satisfied by $d$-dimensional Gaussians with covariance $\Sigma = 4\pi R^2 I$, where $R$ is the size of the truncated space. This still leads to a density that decays exponentially with $d$ away from the mean, in contrast to uniform distributions. There are also some conditions regarding the shape of the distribution, which are detailed in Supplementary Note 5.

With this, we obtain the following theorem. The detailed statement is given in Theorem 8 in Supplementary Note 5.

**Theorem 4.** (Non-uniform distributions; Informal Version of Theorem 8 in Supplementary Note 5) Let $\epsilon > 0$, and let $\varphi^2$ be a sufficiently flat distribution. Let $g_{w^\star} : \mathbb{R}^d \to [-1, 1] \in \mathcal{C}$ be a target function for an unknown vector $w^\star \in \mathbb{R}^d$ with norm $R_w$. Then, there exists a quantum algorithm with QSQ access to a suitably discretized quantum example

state $|g_{w^*}\rangle$ that can efficiently find parameters $\hat{\beta} \in \mathbb{R}^D$ such that $\mathcal{L}_{w^*}(\hat{\beta}) \le \epsilon$ with high probability using

$$N = \mathcal{O}(dD\,\mathrm{polylog}(d, D, R_w, 1/\epsilon)) \tag{6}$$

QSQs and $t = \Theta(\log(D/\epsilon))$ iterations of gradient descent.

As a special case, we obtain the same guarantee for the natural distributions of Gaussians, generalized Gaussians, and logistic distributions. We prove that these distributions are also Fourier-concentrated and hence give us significant quantum advantages, specified in the following corollary.

**Corollary 1.** (Informal) The guarantee of Theorem 4 holds taking $\varphi^2$ as Gaussian, generalized Gaussian, or logistic distributions with large enough scale parameters. Meanwhile, any classical gradient-based algorithm requires

- $\exp(\Omega(\min(d, R_w^2)))$ samples for Gaussian distributions.
- $\Omega(\min(\exp(d), \mathrm{superpoly}(R_w)))$ samples for generalized Gaussian distributions.
- $\exp(\Omega(dR_w))$ samples for logistic distributions.

This is a direct implication of the previous theorem combined with Propositions 2 and 3 and Corollary 9 in Supplementary Note 5. Thus, we see that for Gaussian, generalized Gaussian, and logistic distributions, we obtain an exponential quantum advantage over classical gradient methods. For Gaussian distributions, we retain this exponential advantage over correlational SQ algorithms.

Our key observation is that the classical hardness of[82] stems from the objective function $\mathcal{L}_{w^*}$ being sparse in Fourier space. This implies that the objective function is difficult to optimize using gradient-based methods. On the other hand, quantum algorithms can typically take advantage of Fourier-sparsity by leveraging the quantum Fourier transform (QFT). In fact, we notice that the target functions $g_{w^*}$ are periodic in each coordinate with period $1/w_j^*$:

$$g_{w^*}\left(x + \frac{e_j}{w_j^*}\right) = \tilde{g}\left(\left(x + \frac{e_j}{w_j^*}\right)^\top w^*\right) = \tilde{g}(x^\top w^* + 1) = \tilde{g}(x^\top w^*) = g_{w^*}(x), \tag{7}$$

where we use that $\tilde{g}$ has period 1 and use $e_j$ to denote the unit vector for coordinate $j \in [d]$. Thus, information about the unknown vector $w^*$ is contained in the period of $g_{w^*}$. This observation yields a simple quantum algorithm: (1) Perform period finding by encoding the QFT into QSQs to learn the vector $w^*$ one component at a time, (2) Learn the unknown parameters $\beta_j^*$ defining the periodic activation function ((3)) using classical gradient methods. Note that once we have an approximation of $w^*$ from Step (1), Step (2) is effectively a regression problem, allowing it to be solved via gradient methods.

Despite the initial simplicity of this algorithm, there are several nontrivial issues that arise, particularly in Step (1). First, recall that the quantum example state ((1)) must be suitably discretized because our target function is real. However, there exist pathological examples in which discretization eliminates any information about the period of the original function (see, e.g., Section 10 of ref. 115). Thus, it is important to choose the correct discretization such that the period is sufficiently preserved. Another problem is that the standard period finding algorithm does not apply because the period $1/w_j^*$ is not necessarily an integer. Additionally, standard period finding is only analyzed for uniform superpositions, whereas we are primarily interested in non-uniform superpositions.

To resolve these problems, we carefully discretize the target function such that it satisfies *pseudoperiodicity*[116] with a period proportional to $1/w_j^*$ in each coordinate. For a period $S$, instead of requiring that $h(k) = h(k + \ell S)$ for an integer $\ell$, pseudoperiodicity dictates that $h(k) = h(k + [\ell S])$, where $[\ell S]$ denotes rounding $\ell S$ either up or

down to the nearest integer. This ensures that the period of the discretized function still contains useful information, thus excluding pathological discretizations. Then, for uniform distributions, we can use Hallgren's algorithm[116], which finds the (potentially irrational) period of pseudoperiodic functions. It is still nontrivial to apply Hallgren's algorithm, as it crucially assumes the existence of an efficient verification subroutine to check if a given guess is close to the period of a pseudoperiodic function. Unlike for periodic functions, such verification is not straightforward for pseudoperiodic functions. We design a suitable verification procedure which uses $D$ QSQs in Theorems 6 and 10 in Supplementary Notes 4A2 and 5A2, respectively.

Moreover, Hallgren's algorithm does not apply for non-uniform distributions. To this end, we design a new period finding algorithm that works for sufficiently flat non-uniform distributions, which could be of independent interest. The sufficiently flat condition on the distributions stems from our generalization of Hallgren's algorithm as well as several integral bounds needed for Step (2) of the algorithm. We expand on these ideas in the Methods and Supplementary Notes 4 and 5.

## Discussion

Numerous works have shown exponential quantum advantages for learning Boolean functions when the input data is uniformly distributed. However, little is known about distributions other than uniform, and settings in classical ML commonly consider real-valued functions, leaving a large gap between known quantum advantages and classical ML in practice. Our work makes significant progress towards understanding quantum advantage for learning real functions over non-uniform distributions. Moreover, the function class of periodic neurons that we consider is well-studied in the deep learning theory literature.

One question that has persisted around many quantum learning results, including the present work, is the practical origin of the quantum example state $|c^*\rangle$ for a target function $c^*$. Creating an example state is straightforward when efficient classical descriptions of $c^*(x)$ and the distribution $\mathcal{D}(x)$ are known, and some conditions on the distribution $\mathcal{D}$ are satisfied[117–121]. However, by definition of the problem, $c^*$ is unknown and is precisely what we wish to learn. Instead, one may consider coherently loading the data from known classical examples, but this can be costly and eliminate an end-to-end quantum advantage. For instance, the spacetime volume of the loading circuit is likely to scale exponentially in $d$[122,123], erasing any practical advantage.

Alternatively, we consider the following perspective on how learning may still be valuable even when a description of $c^*$ is known. Suppose we know some complex classical circuit/function that simulates classical physics. One may instead hope to learn a simpler circuit that can approximately compute the same dynamics more efficiently (e.g., ref. 124). Here, the simpler circuit is unknown to the learner, but the algorithm has access to it through the known, complicated circuit that simulates the same dynamics. In this case, because $c^*$ is known, one can construct the quantum example state straightforwardly (albeit with some overhead), which may be helpful in learning a simpler description of the target.

As a practically-relevant example in ML, one can consider the complicated object as a trained neural network. Such models are highly complex, and while there are heuristic methods for constructing them, there is limited understanding of how neural networks compute their outputs. As in the subfield of interpretability in ML[125–127], one may hope to use quantum access to the (known) trained neural network to extract information about this complex model. By representing it differently or learning a simpler model that performs approximately the same function, this could help us better understand opaque large ML models and, in turn, design better ones using our new knowledge of their inner workings. One may object that it is not clear if the type of structure that quantum algorithms typically leverage to

obtain advantages are present in this setting. However, we note that there is some evidence of periodic structure in the features of large language models[128]. Regardless, we hope that this perspective provides new insight into scenarios when quantum example states may occur naturally and be efficiently preparable.

Our work also raises many interesting open questions. First, the classical hardness for our results only holds against classical gradient methods and correlational SQ algorithms (for Gaussian distributions). While there are results proving hardness for classical SQ algorithms or even general classical algorithms[83,84], these results do not directly apply to our parameter regimes. Can the classical hardness be strengthened for our setting? We expect the hardness to still hold and leave this generalization to future work.

Second, we assume that the periodic neuron takes a specific form given by $\tilde{g}$. Could our results be generalized to apply for any periodic function? In addition, while our results hold for a broad class of non-uniform distributions including Gaussians, generalized Gaussians, and logistic distributions, one may wonder if similar results can be obtained for other natural non-uniform distributions. We conjecture that our results could be modified to apply to generalized logistic distributions[129] or stable distributions[130]. It is also possible that the conditions needed for the non-uniform distributions we consider, i.e., the sufficiently flat condition, could be relaxed, although this would require a significantly different analysis. More generally, can one obtain a quantum advantage for this task when learning over any Fourier-concentrated distribution? The main part of the proof that requires modification is the analysis of the non-uniform period finding algorithm.

Finally, while we consider quantum access to real functions via discretized quantum example states, one may consider alternative models for learning classical functions encoded in quantum states, e.g., continuous variable states. Would different models provide new capabilities for quantum learning algorithms?

## Methods
### Classical hardness

In this section, we give an overview of the proofs of Theorems 1 and 2. First, to prove Theorems 1, we adapt the proof of Theorem 4 from[82] to hold for our concept class $\mathcal{C}$, which imposes additional constraints on the vector $w^\star$. Namely, for our setting, $w^\star$ is restricted to the positive orthant and bounded away from 0; meanwhile, in ref. 82, $w^\star$ can be any $d$-dimensional vector with norm $R_w$. We must argue that these restrictions do not make the problem easier for classical algorithms.

The crux of the argument of[82] shows that for any function $h$, for a random choice of $w^\star$, the Fourier transform of the target function $g_{w^\star}$ does not correlate well with $h$ (see Lemma 1 in Supplementary Note 3A). Thus, no matter what our hypothesis function is, obtaining information about $w^\star$ is difficult. However, their proof crucially uses that $w^\star$ is chosen randomly from an exponentially large set of nearly orthogonal vectors, and their construction of this set does not adhere to our requirements for $w^\star$. Our main technical contribution for this proof is showing the existence of a large set $\tilde{S}_w$ of nearly orthogonal vectors which lie in the positive orthant and are bounded away from zero. We do so by leveraging tools from high-dimensional geometry[131].

In Theorem 2, we strengthen the classical hardness to hold against any algorithms using correlational SQs when learning with respect to Gaussian distributions. A standard method for proving correlational SQ lower bounds is via the *statistical dimension*[95,110,114], which captures the difficulty of learning a concept class, similarly to the more commonly known VC dimension. Informally, the statistical dimension quantifies the size of the largest subset of the concept class whose elements have "low correlation" (see Definition 4 in Supplementary Note 3B for a formal definition). Intuitively, functions in this low correlation subset should be hard to distinguish and thus hard to learn. Several works[114,132,133] have formalized this relationship, proving,

roughly, that lower bounds on the statistical dimension imply correlational SQ lower bounds (see Theorem 3 in Supplementary Note 3B). Our key contribution is proving an exponential lower bound on the statistical dimension of our concept class $\mathcal{C}$, where we utilize properties of Gaussian integrals and the construction of the set $\tilde{S}_w$ from the proof of Theorem 1 described above. Thus, this results in an exponential lower bound for correlational SQ algorithms.

### Quantum algorithm

In this section, we describe the ideas behind the proofs of Theorems 3 and 4. As discussed above, our key observation is that the target function $g_{w^\star}$ is periodic in each coordinate with period $1/w_j^\star$. This informs our quantum algorithm, which is as follows: (1) Perform period finding by encoding the QFT into QSQs to learn the vector $w^\star$ one component at a time; (2) Learn the unknown parameters $\beta_j^\star$ defining the periodic activation function ((3)) using classical gradient methods. Thus, the proofs are separated into two main parts, each analyzing the sample complexities for these two algorithmic steps. The proofs for the uniform distribution are in Supplementary Note 4, and those for the non-uniform distributions are in Supplementary Note 5. In particular, Step (1) is analyzed in detail in Supplementary Notes 4A and 5A, and Step (2) is examined in Supplementary Notes 4B and 5B. In the following, we give an overview of the proofs.

**Learning the linear function.** We want to apply period finding to our target function $g_{w^\star} : \mathbb{R}^d \to [-1, 1]$ to approximate the unknown vector $w^\star$ that defines the inner linear function. First, because $g_{w^\star}$ has real inputs and outputs, we need to discretize it so that it can be represented by a (discrete) quantum example state ((1)). We require that the chosen discretization is pseudoperiodic, a condition which is weaker than periodicity but still ensures that the discretized function retains information about the period of $g_{w^\star}$. Specifically, for $d = 1$, a function $h : \mathbb{Z} \to \mathbb{R}$ is pseudoperiodic with period $S \in \mathbb{R}$ if $h(k) = h(k + [\ell S])$ for any integer $\ell$, where $[\ell S]$ denotes rounding $\ell S$ either up or down to the nearest integer. One should compare this to periodicity, where the necessary condition is instead $h(k) = h(k + \ell S)$. We choose the following discretization, considering $d = 1$ for simplicity. We generalize to arbitrary $d \geq 1$ in the discussion surrounding Supplementary Equation (254).

**Lemma 1.** (Discretization; Informal) Let $M_1$, $M_2$ be suitably chosen discretization parameters with $M_1 > M_2$. Consider the discretized function $h_{M_1, M_2} : \mathbb{Z} \to \frac{1}{M_2} \mathbb{Z}$ defined by

$$h_{M_1, M_2}(k) \triangleq \left\lfloor g_{w^\star}\left(\frac{k}{M_1}\right) \right\rfloor_{M_2}, \tag{8}$$

where $\lfloor \cdot \rfloor_{M_2}$ denotes rounding down to the nearest multiple of $1/M_2$. Then, $h_{M_1, M_2}$ is pseudoperiodic with period $M_1/w^\star$ for a large proportion of the inputs.

Requiring $M_1 > M_2$ at an appropriate ratio makes the discretization more coarse on the outputs than the inputs, ensuring that pseudoperiodicity is satisfied. In our proof, we choose $M_1, M_2$ to scale polynomially in the problem parameters, i.e., poly($\epsilon, D, d, R_w$), where $\epsilon$ is the desired error, $D$ is the number of cosine terms in the periodic activation function ((3)), $d$ is the input dimension, and $R_w$ the norm of $w^\star$.

Recall that $g_{w^\star}$ has period $1/w_j^\star$ in the $j$th coordinate. Thus, learning the period of $h_{M_1, M_2}$ also allows us to approximate the period of $g_{w^\star}$. However, straightforwardly applying standard period finding algorithms to $h_{M_1, M_2}$ fails because $h_{M_1, M_2}$ is only pseudoperiodic rather than periodic and its period $M_1/w_j^\star$ is not necessarily an integer. Instead, we turn to Hallgren's algorithm[116], which determines the period of pseudoperiodic functions and applies to real periods. Note that Hallgren's algorithm only applies for the uniform distribution, so we consider this case for now. At a high level, Hallgren's algorithm first quantum Fourier samples twice and computes the continued fraction

expansion of the quotient of the results. Then, it constructs a guess for the period for each convergent of the expansion and iterates through each guess, checking which one approximates the period. Ref. 116 shows that one guess is guaranteed to be close to the period. We discuss Hallgren's algorithm in more detail in Supplementary Note 1B.

Notice that a crucial subroutine necessary for Hallgren's algorithm is a verification procedure to check if a given guess is close to the period of a pseudoperiodic function. Unlike for periodic functions, where such verification is straightforward, this is nontrivial for pseudoperiodic functions. In fact, ref. 116 leaves this as an assumption to be instantiated upon applying the guarantee of Hallgren's algorithm.

We design a suitable verification procedure which uses $D$ QSQs (see Theorems 6 and 10 in Supplementary Notes 4A2 and 5A2, respectively). The main idea is to compute the inner product between $h_{M_1, M_2}$ and $h_{M_1, M_2}(\cdot + T)$, where $T$ is a guess for the period. Intuitively, this inner product should be large for a guess that approximates the period well. We define an observable that allows us to compute this inner product using QSQs. Then, we identify a suitable threshold which the inner product surpasses if and only if the guess is indeed close to the period. The majority of the technical work for the verification procedure lies in finding such a threshold. With this, the only remaining quantum part of Hallgren's algorithm is quantum Fourier sampling, which can be accomplished using QSQs by encoding the QFT into the queried observable. Because we need to repeat this algorithm for each entry in the vector $w^\star \in \mathbb{R}^d$, we use $\tilde{\mathcal{O}}(dD)$ QSQs, where the polylogarithmic factors come from amplifying the success probability of Hallgren's algorithm.

Thus far, we discussed how to utilize Hallgren's algorithm for our problem, which only applies for uniform distributions. We generalize these ideas to perform period finding for non-uniform input distributions. This algorithm can be found explicitly in the appendices in Algorithm 3 in Supplementary Note 5B2, and the verification procedure is presented in Algorithm 4 in Supplementary Note 5B2. Our algorithm follows the same structure as Hallgren's algorithm but requires a new analysis due to the different input distribution. Here, we crucially use that the non-uniform distributions we consider are sufficiently flat, e.g., they are pointwise-close to uniform. As discussed previously, our flatness condition only requires the univariate (unnormalized) marginals to be close to uniform, but the overall density can decay exponentially in $d$.

**Learning the periodic activation function.** In the previous section, we showed how to obtain an approximation $\hat{w}$ of the unknown vector $w^\star$ using quantum period finding. Using this approximation, we can learn the unknown parameters $\beta_j^\star$, which determines the periodic activation function $\tilde{g}$ given in (3). This step of the algorithm is purely classical.

With the approximation $\hat{w}$, we can consider predictors $f_\beta$ defined by

$$f_\beta(x) \triangleq \sum_{j=1}^{D} \beta_j \cos(2\pi j x^\top \hat{w}), \tag{9}$$

where $\beta \in \mathbb{R}^d$ is a vector of trainable parameters. These predictors have the same form as the target function $g_{w^\star}$ but replace $w^\star$ and $\beta_j^\star$ with $\hat{w}$ and $\beta_j$, respectively. Thus, the loss function from (4) can be written more explicitly as

$$\mathcal{L}_{w^\star}(\beta) = \int_{x \sim \varphi^2} \left( \sum_{j=1}^{D} \beta_j^\star \cos(2\pi j x^\top w^\star) - \sum_{j=1}^{D} \beta_j \cos(2\pi j x^\top \hat{w}) \right)^2 dx. \tag{10}$$

We use (approximate) gradient access to this loss function to learn parameters $\hat{\beta}$ such that $\mathcal{L}_{w^\star}(\hat{\beta}) \leq \epsilon$. We acknowledge there may be other

approaches to solve for the parameters, but we believe gradient descent is the most straightforward. First, we show that the gradients are informative, i.e., the derivative of the objective function $\partial \mathcal{L}_{w^\star} / \partial \beta_k$ indeed reflects how far $\beta_k$ is from the true parameter $\beta_k^\star$. With this, we can simply apply gradient descent (see, e.g., ref. 134), where we show that the iterates converge to the true parameters within $t = \Theta(\log(D/\epsilon))$ steps. Most of the work in this step goes into carefully choosing the hyperparameters (e.g., the number of iterations to run gradient descent, how accurate the approximation $\hat{w}$ is required to be, etc.) to guarantee that the value of the loss function is small. In this step, the proofs for uniform and non-uniform distributions are very similar. The full proofs are provided in Supplementary Note 4B for the uniform case and Supplementary Note 5B for the non-uniform case.

## Data availability

No data are generated or analyzed in this theoretical work.

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

## Acknowledgements

The authors thank Andrew Childs, András Gilyén, Hsin-Yuan (Robert) Huang, Robbie King, Robin Kothari, and Chirag Wadhwa for helpful discussions. L.L. was supported by a Marshall Scholarship. This work was done (in part) while a subset of the authors were visiting the Simons Institute for the Theory of Computing.

## Author contributions

D.G. and J.R.M. conceived the project. L.L. and D.G. developed the mathematical aspects of the work. All authors contributed to the writing of the manuscript.

## Competing interests

The authors declare no competing interests.
