## [Transparent Peer Review file · Nature Communications]

Quantum advantage for learning shallow neural networks with natural data distributions

Corresponding Author: Ms Laura Lewis

Version 0:

Reviewer comments:

Reviewer #1

(Remarks to the Author)

The ultimate goal of Quantum Machine Learning (QML) is to achieve practical quantum advantages in data analysis — which is by no means a trivial feat. Due to the lack of good quality large-scale quantum hardware, theoretical studies are needed to guide the path, in particular rigorous demonstration of a quantum advantage in a more analytically manageable (despite not reflecting the real world problem) setting.

The work makes a novel contribution by attempting to show the advantage in the Quantum Statistical Query (QSQ) framework for learning periodic neurons (i.e., a linear combination of cosine functions which take an output from a single layer of neural networks as an argument) over non-uniform input distributions. In particular, there are two key improvements upon prior results in the QSQ framework with a quantum example state: (1). The target functions are real-valued together with input data (instead of, Boolean and bit-string, respectively, commonly used in this framework), which means the authors are required to come up with an appropriate choice of discretization for a quantum example state in Eq. (II.1) for the QSQ framework, and (2). The input data distributions do not need to be uniform as long as they are “sufficiently flat” — with some provided examples including (generalized) Gaussian and logistic distributions. The developed QSQ-based algorithm consists of two main steps: (1). Period finding which employs the variant of the Quantum Fourier Transform to find relevant periods of the target functions using the quantum example state, and (2). Solving regression with gradient descents which classically determines the coefficients associated with individual cosine functions.

I did enjoy reading the work and think the improved version would significantly contribute to the field of QML. However, I cannot yet make a recommendation for publication without the following comments/concerns being addressed.

(1). My biggest concern is that to properly claim the quantum advantage, it is necessary to prove the classical hardness of learning periodic neurons beyond gradient descent methods. I appreciate I may not be asking for an easy task, but given that training the loss function suffers from a classical version of barren plateau there might be the way to extend the current proof of the classical hardness with the gradient-based methods.

(2). The notion of Fourier concentration should be elaborated more in the main text, since it plays a central role on the classical (gradient-based) hardness. For example, the infinite sum of $\epsilon(n R\omega/4)$ over n in Theorem 1 should be explained that it does not diverge.

(3). The hardness statements of Theorem 1 and Corollary 1 are not stated in a consistent manner e.g., in Corollary 1, it is stated in terms of exponential sample scaling (which is more preferable personally, but happy with any version as long as it is consistent).

(4). Regarding Corollary 1, if we choose $R\omega \in O(1)$ since d scales at most polynomially with the number of qubits, the lower bounds on the sample complexity seem to be too loose for (generalized) Gaussian distributions to claim the classical hardness. Do we need an additional assumption on $R\omega$ for this ?

(5). It should be noted that barren plateaus which are the root of classical hardness here are the classical version and fundamentally differ from the quantum version.

(6). The condition on the “sufficiently flat” distribution needs to be explained in more details in the main text. In particular, this condition does not appear on the sample complexity of the current Theorem 3. Additionally, this part of the sentence “[...] yet not so large that the distribution is effectively indistinguishable from uniform.” may need to be revisited — what does it mean to be “effectively indistinguishable from uniform” here ? More minor comment is that the notion of truncated space seems appear a bit out of no where.

(7). The proof of the quantum learnability relies sufficiently on the sufficient flatness condition, which is stronger than the Fourier concentrated condition (as explained in Appendix). Maybe it is worth stating what those additional assumptions play a role in the proof — elaborating on this also enhances the closing sentence in the hope of extending the advantage result more generally with only the Fourier concentration.

(8). The current title is a bit misleading with “natural data distribution” ? — Though, it’s true that these non-uniform distributions particularly Gaussian appear frequent in nature. However, I am not sure if I agree that these distributions represent the nature of how real-world data are distributed ? Furthermore, the notion of “periodicity” which is crucial to both quantum learnability and classical hardness is also missing in the title.

(9). "To this end, we design a new period finding algorithm that works for sufficiently flat non-uniform distributions, which could be of independent interest." — I tend to agree with the authors on this, which is why I feel like this aspect should be elaborated. With such a feat and important key aspect of the algorithm, the technical details are a bit underrepresented. For example, maybe it is worth to include this developed period finding algorithm explicitly ?

(10). Regarding the "solving regression with gradient descents" part, the analysis/proof does not seem to take into account the finite sample effect of training data. In particular, Eq (V3) averages over all the input space with respect to the considered input distribution. What happens to the results when we have instead finite training samples drawn from such input distribution (— the situation which may be more practical) ?

(11). Following from (10), it also occurs to me that these coefficients could be estimated with other optimization approaches. For example, if we have the training dataset of size N such that $N > D$, these coefficients could be obtained with simply solving linear system of equations ?

(12). The discussion on the new aspect of having the known target function and creating the quantum example state for learning a more compact representation of the function is insightful. While it is true that some periodic structure appears relevant in practice for data analysis (— this together with the previous sentence giving more justification of the quantum example from a different angle), I do not think this specific setting considered in this work is practically relevant and honestly I think this is fine.

(13). “Are there classes of real-valued functions and non-uniform distributions for which quantum data is advantageous?” — while it is true that this work is the first to positively answer this with the quantum example state, Molteni et al [arXiv:2405.02027 (2024)] has shown this also in the setting which can be considered to be under the QSQ framework "without" the quantum example state. In that work, the input data distribution is "arbitrary" and the target function is real-valued.

(14). Font size in the Fig 1 diagram is unnecessarily too large.

Reviewer #2

(Remarks to the Author)

Summary:

The submission “Quantum advantage for learning shallow neural networks with natural data distributions” investigates the query complexity of learning functions that are a composition of a periodic function and a linear function (aka periodic neurons) in the distribution specific setting, given access to a quantum statistical query (QSQ) oracle that can output noisy expectation values of the quantum example state. The loss function considered is the expectation of the error between the predictor and the true concept. The main result of the paper is an exponential separation in the query complexity of learning under sufficiently flat distributions with queries from the QSQ oracle and the samples required during any classical gradient-based algorithm.

Evaluation:

There has been a lot of attention from the quantum learning theory community to separations between learning classes of Boolean functions from random examples classically and using the quantum example state (or qsamples) quantumly. This work seems a natural extension of prior work in considering possible separations between learning real-valued functions using a classical gradient based algorithm using samples versus a quantum algorithm using a QSQ oracle.

From a technical perspective:

- Appropriate assumptions: Regarding the quantum example state, choices were made in the discretization of the real-valued function being encoded to ensure information regarding the periodicity was retained, which the authors call pseudoperiodicity.
- Hallgren’s algorithm: In order to apply Hallgren’s algorithm for finding the period of the function, the authors have to implement a verification protocol that makes few queries to the QSQ oracle to test if a given guess is close to the period.
- Difficulty for classical-gradient algorithms: The authors design lower bounds for the learning task for classical-gradient

algorithms by building on Ref. [80].

Comments regarding the paper:

- At the end of section II “Problem Definition”, it is remarked that having access to the derivative is more restrictive than general classical SQ algorithms. Why are classical gradient algorithms then best suited for this task and the best classical baseline for the quantum learner?
- Could the authors also discuss why the SQ setting is a natural generalization of the gradient access model in a bit more detail? This does not seem substantiated currently.
- I found the sections of “Introduction” and “Problem Definition” well-written even for an audience outside of this subarea.
- In lines 57-60, the authors remark that Ref. [80] mention that periodic neurons cannot be learned efficiently under distributions with sparse Fourier transform. At first glance, I would have thought that sparsity of the Fourier spectrum would help. Could you please provide some intuition why this is not the case?
- In lines 87-88, the authors remark that the quantum example state (II.1) is not known to be efficiently preparable for all distributions but I could not find a discussion regarding this later except for in lines 1783-1784 but only for $d=1$. What about higher values of d that this paper seems to mainly care about?
- From looking at Corollary 10, it appears to me that you also have an exponential separation in D (and not just d) as long as the first term of (E.7) is appropriately bounded. Perhaps, this should be mentioned as part of Corollary 1 to indicate that there is an advantage even for $d=1$ for certain regimes. (I might be mistaken here. I did not go through the proof of Corollary 10 completely.)

Version 1:

Reviewer comments:

Reviewer #1

(Remarks to the Author)

I would like to thank the authors for the revised manuscript. The authors have addressed most of my previous comments. In particular, the newly added Theorem 2 strengthens the claim of quantum advantage by extending the analysis to a family of correlational SQs.

There are, however, a few remaining points that should be addressed before I can recommend the manuscript for publication.

- (1). Proof sketch of Theorem 2: A brief proof sketch of Theorem 2 should be added to the Methods section of the main text. The authors have used a distinct approach to establish this result, and I believe it would greatly benefit readers to include a concise overview of the underlying reasoning.
- (2). Figure 1 revision: Figure 1 should be revised to reflect the setting of Theorem 2. Specifically, incorporating elements related to momentum/etc should suffice.
- (3). Discussion of related literature: I strongly recommend that the authors broaden their discussion of the quantum machine learning literature to include other aspects and clarify how their work connects to prior studies. In particular, the following works should be addressed:
 - (3.1). Regarding Molteni et al., arXiv:2405.02027 (2024), I agree with the authors’ response that the settings differ. In addition, I think the underlying hardness assumptions are distinct. Nevertheless, it would be helpful for readers if the authors explicitly highlight the distinctions between the two works.
 - (3.2). It would also be valuable for the authors to discuss how their results relate to the fundamental perspective of quantum barren plateaus, for example as in [Larocca et al., Nature Reviews Physics, 1–16 (2025)] for the general review and [Nietner, arXiv:2310.17716 (2023)] for specific connection to SQ-learning.
- (4). In response to the authors’ answer to my point (11) in the previous round, I think the authors could be more precise about how alternative approaches (i.e., methods other than gradient-based optimization) can be used to solve for the coefficients in Eq. (V3). In practice we only have access to a finite set of training data, so we work with an empirical version of Eq. (V3). In that setting, could one simply apply a linear-regression approach (e.g., Ridge/LASSO) to estimate these coefficients when $N > D$?

Reviewer #2

(Remarks to the Author)

Thank you for the detailed responses. I am happy with the edits made.

Reviewers of *Nature Communications*

Dear Reviewers,

We thank you for your detailed reading of our work and your suggestions to improve its suitability for the broad readership of *Nature Communications*. We have addressed your comments below and have implemented the corresponding changes in the manuscript. Edits/deletions to the original text are highlighted in red and additions are made in blue. Thank you for your time in considering our work.

Sincerely,
Laura Lewis, Dar Gilboa, and Jarrod R. McClean

Reviewer #1 (Remarks to the Author):

The ultimate goal of Quantum Machine Learning (QML) is to achieve practical quantum advantages in data analysis — which is by no mean a trivial feat. Due to the lack of good quality large-scale quantum hardware, theoretical studies are needed to guide the path, in particular rigorous demonstration of a quantum advantage in a more analytically manageable (despite not reflecting the real world problem) setting.

The work makes a novel contribution by attempting to show the advantage in the Quantum Statistical Query (QSQ) framework for learning periodic neurons (i.e., a linear combination of cosine functions which take an output from a single layer of neural networks as an argument) over non-uniform input distributions. In particular, there are two key improvements upon prior results in the QSQ framework with a quantum example state: (1). The target functions are real-valued together with input data (instead of, Boolean and bit-string, respectively, commonly used in this framework), which means the authors are required to come up with an appropriate choice of discretization for a quantum example state in Eq. (II.1) for the QSQ framework, and (2). The input data distributions do not need to be uniform as long as they are “sufficiently flat” — with some provided examples including (generalized) Gaussian and logistic distributions. The developed QSQ-based algorithm consists of two main steps: (1). Period finding which employs the variant of the Quantum Fourier Transform to find relevant periods of the target functions using the quantum example state, and (2). Solving regression with gradient descents which classically determines the coefficients associated with individual cosine functions.

I did enjoy reading the work and think the improved version would significantly contribute to the field of QML. However, I cannot yet make a recommendation for publication without the following comments/concerns being addressed.

Author response: We thank the reviewer for their accurate summary and close reading of our work. We have endeavored to address all comments below and made corresponding changes in the manuscript.

(1). *My biggest concern is that to properly claim the quantum advantage, it is necessary to prove the classical hardness of learning periodic neurons beyond gradient descent methods. I appreciate I may not be asking for an easy task, but given that training the loss function suffers from a classical version of barren plateau there might be the way to extend the current proof of the classical hardness with the gradient-based methods.*

Author response: We thank the reviewer for their comment and agree that stronger classical hardness would enhance our claim of quantum advantage, and believe we have now strengthened our result. Nevertheless, we emphasize that gradient methods are definitively the most common algorithm used in practice to learn functions of this form. Moreover, because the classical hardness stems from Fourier sparsity, we do not expect any classical algorithm to overcome this barrier. Thus, our current proof already provides strong evidence of general classical intractability. Additionally, we again emphasize that results in the classical literature provide strong evidence for more general hardness, see Refs. [84, 85].

However, we acknowledge the reviewer’s perspective and have strengthened the classical hardness to hold against *correlational* statistical queries (SQs). This class is more general than gradient methods, as correlational SQs also include dimension reduction and moment-based methods (albeit they are still restricted compared to general classical SQs, which would be the most equal comparison to our quantum access model). We state this new result in Theorem 2 in the revised main text, and the proof is given in Appendix C.2. To prove it, we take a different approach from our gradient hardness result. Instead, we prove an exponential lower bound the statistical dimension (the analogue of VC dimension for SQ algorithms) of our concept class, which imposes a fundamental barrier against efficient learners. We have also edited the main text to discuss the result further. Given the length of the current paper and the short review period, we leave further extensions of the classical hardness to future work.

(2). *The notion of Fourier concentration should be elaborated more in the main text, since it plays a central role on the classical (gradient-based) hardness. For example, the infinite sum of $\epsilon(nR_w/4)$ over n in Theorem 1 should be explained that it does not diverge.*

Author response: We thank the reviewer for pointing this out. We note that if the infinite sum indeed diverges, then the sample complexity lower bound is vacuous. Thus, one can view it as imposing a condition on the input distribution (e.g., being Fourier concentrated with at least super-polynomially decaying tails) to obtain a meaningful lower bound. We have clarified this in the text below the theorem statement.

(3). *The hardness statements of Theorem 1 and Corollary 1 are not stated in a consistent manner e.g., in Corollary 1, it is stated in terms of exponential sample scaling (which is more preferable personally, but happy with any version as long as it is consistent).*

Author response: We appreciate this feedback and have edited the statement to be more consistent with Corollary 1. However, we clarify that Theorem 1 is more general than Corollary 1 and will only imply specific query complexity lower bounds for particular choices of distributions. We state it as such for consistency with the classical literature and to emphasize that classical hardness holds for more general distributions than the examples we give.

(4). *Regarding Corollary 1, if we choose $R_w \in O(1)$ since d scales at most polynomially with the number of qubits, the lower bounds on the sample complexity seem to be too loose for (generalized) Gaussian distributions to claim the classical hardness. Do we need an additional assumption on R_w for this?*

Author response: Thank you for the question. Both R_w and d should be considered as large

parameters. This is the same setting as for gradient-based hardness (Ref. [80]).

(5). *It should be noted that barren plateaus which are the root of classical hardness here are the classical version and fundamentally differ from the quantum version.*

Author response: We thank the reviewer for their comment and have explained how quantum algorithms overcome the source of classical hardness in the text below Theorem 1. We also think that the use of the term “barren plateau” may confuse readers due to its association with barren plateaus in variational QML (as suggested by the reviewer) and have removed it from the text. Instead, we describe the classical hardness as stemming from the gradient of the loss function being extremely concentrated, which is also more precise.

(6). *The condition on the “sufficiently flat” distribution needs to be explained in more details in the main text. In particular, this condition does not appear on the sample complexity of the current Theorem 3. Additionally, this part of the sentence “[...] yet not so large that the distribution is effectively indistinguishable from uniform.” may need to be revisited — what does it mean to be “effectively indistinguishable from uniform” here? More minor comment is that the notion of truncated space seems appear a bit out of no where.*

Author response: We apologize for the lack of clarity. The sufficiently flat condition does not need to appear in the sample complexity. Any parameters introduced by this condition are constants and thus absorbed into the big-oh in the sample complexity, so they will not appear explicitly. We also thank the reviewer for pointing out the previously un-introduced notion of a truncated space and have added a description to the end of Section II.

By the informal phrase “effectively indistinguishable from uniform,” we meant to convey that our algorithm is able to tolerate significant deviations of the data distribution from uniform with respect to, e.g., ℓ^∞ distance. The ability to handle such deviations stems from the element-wise nature of our algorithm. We have changed the wording of this phrase in the manuscript.

(7). *The proof of the quantum learnability relies sufficiently on the sufficient flatness condition, which is stronger than the Fourier concentrated condition (as explained in Appendix). Maybe it is worth stating what those additional assumptions play a role in the proof — elaborating on this also enhances the closing sentence in the hope of extending the advantage result more generally with only the Fourier concentration.*

Author response: We thank the reviewer for their suggestion. We previously explained the role of the sufficiently flat condition at the end of Section III and the end of Section V.A, namely that it is important for the analysis of our non-uniform period finding algorithm. As suggested, we have also added a sentence about this in the discussion near the open question the reviewer mentions.

(8). *The current title is a bit misleading with “natural data distribution”? — Though, it’s true that these non-uniform distributions particularly Gaussian appear frequent in nature. However, I am not sure if I agree that these distributions represent the nature of how real-world data are distributed? Furthermore, the notion of “periodicity” which is crucial to both quantum learnability and classical hardness is also missing in the title.*

Author response: We thank the reviewer for their perspective on this, but, after much consideration, we would prefer to keep our current title. We refer to the non-uniform distributions we consider as “natural” due to the classical literature, e.g., Ref. [80]. Regarding periodicity, while our quantum algorithm indeed relies on this property, classical hardness holds for a variety of different activation functions (see, e.g., Ref. [82]). We think that “shallow neural networks” in the title makes the work more accessible to a broad audience, rather than a more technical description

such as “periodic neurons.” This also emphasizes that the functions we consider form the basic structure of neural networks with periodic activation functions, which are well-studied in classical ML (see, e.g., Refs. [84-89]). Nevertheless, we mention periodicity in the abstract and throughout the work.

(9). *“To this end, we design a new period finding algorithm that works for sufficiently flat non-uniform distributions, which could be of independent interest.” — I tend to agree with the authors on this, which is why I feel like this aspect should be elaborated. With such a feat and important key aspect of the algorithm, the technical details are a bit underrepresented. For example, maybe it is worth to include this developed period finding algorithm explicitly?*

Author response: Thank you for your positive feedback. The ideas for the uniform case are presented in Section V.A, and the non-uniform case is quite similar but just requires additional technical effort (e.g., some of the bounds do not work out as nicely as the uniform case). We thus find it more instructive to present the ideas of the algorithm. However, as suggested, we have added a reference to the explicit algorithm in Section V.A.

(10). *Regarding the “solving regression with gradient descents” part, the analysis/proof does not seem to take into account the finite sample effect of training data. In particular, Eq (V.3) averages over all the input space with respect to the considered input distribution. What happens to the results when we have instead finite training samples drawn from such input distribution (— the situation which may be more practical)?*

Author response: Thank you for your question. While the reviewer is correct that the average in Eq. (V.3) is taken over the entire input space, our access model assumes that we are given access to *approximate* gradients of the loss function in Eq. (V.3). In particular, one could approximate such gradients using a finite amount of labeled training data, as in the case of PAC learning. We have edited the manuscript to make it clearer that we only need approximate access to the gradients. We also note that this is a standard access model/loss function in the classical literature and is not specific to our setting.

(11). *Following from (10), it also occurs to me that these coefficients could be estimated with other optimization approaches. For example, if we have the training dataset of size N such that $N > D$, these coefficients could be obtained with simply solving linear system of equations?*

Author response: Thank you for your thoughtful comments. First, we clarify that having access to labeled training data like in classical PAC learning is a stronger access model than the gradient access we consider. In particular, approximate gradient access is a special case of the statistical query model, which is weaker than having access to PAC examples.

Even if one allows direct access to labeled training data $\{(x, g_{w^*}(x))\}$, then it is not immediately clear if solving a system of linear equations would work. If we understand the reviewer’s suggestion correctly, one could imagine constructing a system of equations in the unknown parameters β_j^* using the known form of g_{w^*} and the values $g_{w^*}(x)$ from the training data. However, from the period finding step of the algorithm, we only have *approximate* knowledge of w^* , so this would only give an approximate system of linear equations, which may be difficult to solve. Nevertheless, we certainly acknowledge that other optimization methods could also be used to find the parameters, and we only use gradient descent since we find it the most straightforward. We have clarified this at the end of Section IV.B.

(12). *The discussion on the new aspect of having the known target function and creating the quantum example state for learning a more compact representation of the function is insightful. While it is true that some periodic structure appears relevant in practice for data analysis (—*

this together with the previous sentence giving more justification of the quantum example from a different angle), I do not think this specific setting considered in this work is practically relevant and honestly I think this is fine.

Author response: We agree with the reviewer in broad terms. In the discussion, we hope to provide a new perspective on some scenarios where one could imagine creating quantum example states since this is generally overlooked in the literature. We agree that our setting is not immediately practically relevant, though we hope these examples can help motivate empirical studies in the future as in classical ML today and thank the reviewer for their positive assessment regardless.

(13). “Are there classes of real-valued functions and non-uniform distributions for which quantum data is advantageous?” — while it is true that this work is the first to positively answer this with the quantum example state, Molteni et al [arXiv:2405.02027 (2024)] has shown this also in the setting which can be considered to be under the QSQ framework “without” the quantum example state. In that work, the input data distribution is “arbitrary” and the target function is real-valued.

Author response: Thank you for this comment. We were aware of this work but did not classify it as addressing the same problem. Namely, we focus on learning functions from classical learning theory encoded in quantum states (e.g., through quantum example states), which is the same setting as the survey we cite for these open questions. In contrast, Molteni et al. considers more quantum-inspired functions arising from expectation values of quantum states. Following the central question, we clarify that “quantum data” refers specifically to the setting of quantum example states considered in this work.

(14). Font size in the Fig 1 diagram is unnecessarily too large.

Author response: We have decreased the font size.

Reviewer #2 (Remarks to the Author):

Summary: The submission “Quantum advantage for learning shallow neural networks with natural data distributions” investigates the query complexity of learning functions that are a composition of a periodic function and a linear function (aka periodic neurons) in the distribution specific setting, given access to a quantum statistical query (QSQ) oracle that can output noisy expectation values of the quantum example state. The loss function considered is the expectation of the error between the predictor and the true concept. The main result of the paper is an exponential separation in the query complexity of learning under sufficiently flat distributions with queries from the QSQ oracle and the samples required during any classical gradient-based algorithm.

Evaluation: There has been a lot of attention from the quantum learning theory community to separations between learning classes of Boolean functions from random examples classically and using the quantum example state (or qsamples) quantumly. This work seems a natural extension of prior work in considering possible separations between learning real-valued functions using a classical gradient based algorithm using samples versus a quantum algorithm using a QSQ oracle.

From a technical perspective: - *Appropriate assumptions:* Regarding the quantum example state, choices were made in the discretization of the real-valued function being encoded to ensure information regarding the periodicity was retained, which the authors call pseudoperiodicity. - *Hallgren’s algorithm:* In order to apply Hallgren’s algorithm for finding the period of the function, the authors have to implement a verification protocol that makes few queries to the QSQ oracle to test if a given guess is close to the period. - *Difficulty for classical-gradient algorithms:* The authors design

lower bounds for the learning task for classical-gradient algorithms by building on Ref. [80].

Author response: We thank the reviewer for their thorough and careful reading of our work.

Comments regarding the paper: - At the end of section II “Problem Definition”, it is remarked that having access to the derivative is more restrictive than general classical SQ algorithms. Why are classical gradient algorithms then best suited for this task and the best classical baseline for the quantum learner?

Author response: Thank you for your comment, we have accordingly further strengthened our hardness result. However, we do believe that gradient algorithms are the best baseline because they are indisputably the most common algorithm used in practice to learn and train neural networks. Thus, the essence of our result is that our quantum algorithm provably outperforms the most practically-relevant class of classical methods used to tackle the problem. Moreover, the proof elucidates that this classical hardness stems from Fourier sparsity, which we do not expect more powerful classical algorithms to be able to overcome. We have added some explanation to the end of Section II.

To address this comment more fully, we have also added a new result which strengthens the classical hardness to hold beyond gradient methods. Namely, we prove that this problem is hard for any algorithm using *correlational* SQs. These are a generalization of gradient methods. In particular, correlational SQs also include dimension reduction and moment-based methods as special cases. We state this new result in Theorem 2 in the revised main text, and the proof is given in Appendix C.2. We have also edited the main text to discuss this result further.

- Could the authors also discuss why the SQ setting is a natural generalization of the gradient access model in a bit more detail? This does not seem substantiated currently.

Author response: Thank you for your valuable feedback. We have expanded on this at the end of Section II. In short, gradient methods are a special case of SQ algorithms (e.g., Ref. [95] which shows that stochastic gradient descent can be simulated using SQs), and the proof of classical hardness for gradient methods is intuitively similar to hardness results in the SQ literature.

- I found the sections of “Introduction” and “Problem Definition” well-written even for an audience outside of this subarea.

Author response: Thank you for your positive comments.

- In lines 57-60, the authors remark that Ref. [80] mention that periodic neurons cannot be learned efficiently under distributions with sparse Fourier transform. At first glance, I would have thought that sparsity of the Fourier spectrum would help. Could you please provide some intuition why this is not the case?

Author response: Thank you for your question. The distribution being sparse in Fourier space implies that the gradient of the loss function is extremely concentrated around a fixed value, independent of what we want to learn (namely, the vector w^*). This means that when one attempts to learn the target function by minimizing the loss function using classical gradient methods, then the gradients will be very flat and thus not helpful in learning the function. This causes one to iterate for a very long time in order to find the solution. Perhaps the reviewer is confused that sparsity in Fourier space should mean there is a smaller space of viable solutions to search through. However, classical algorithms generally cannot search through Fourier space efficiently to identify which elements in Fourier space are nonzero. This is also the source of canonical quantum speedups such as period finding.

- In lines 87-88, the authors remark that the quantum example state (II.1) is not known to be efficiently preparable for all distributions but I could not find a discussion regarding this later except for in lines 1783-1784 but only for $d = 1$. What about higher values of d that this paper seems to mainly care about?

Author response: We apologize for the confusion here. The techniques of Ref. [109] easily generalize to the case of larger d . We had meant that some other references in the lines mentioned only focus on the $d = 1$ case or specific distributions. We have removed the phrase “particularly for the $d = 1$ case” to avoid future confusion.

- From looking at Corollary 10, it appears to me that you also have an exponential separation in D (and not just d) as long as the first term of (E.7) is appropriately bounded. Perhaps, this should be mentioned as part of Corollary 1 to indicate that there is an advantage even for $d=1$ for certain regimes. (I might be mistaken here. I did not go through the proof of Corollary 10 completely.)

Author response: Thank you for the comment. We are not quite clear what the reviewer is suggesting. We clarify that the D parameter does not show up in the classical lower bound. Thus, although our quantum algorithm has only a linear scaling in D , it is difficult to compare to the classical case. Nevertheless, one should really think of D as being a constant since the problem is classically hard even for $D = 1$.

Reviewers of *Nature Communications*

Dear Reviewers,

Thank you again for your helpful comments to improve its suitability for the broad readership of *Nature Communications*. In this revision, we have addressed the remaining comments and implemented the corresponding changes in the manuscript. Edits/deletions to the original text are highlighted in red and additions are made in blue. Thank you for your time in considering our work.

Sincerely,
Laura Lewis, Dar Gilboa, and Jarrod R. McClean

Reviewer #1 (Remarks to the Author):

I would like to thank the authors for the revised manuscript. The authors have addressed most of my previous comments. In particular, the newly added Theorem 2 strengthens the claim of quantum advantage by extending the analysis to a family of correlational SQs.

There are, however, a few remaining points that should be addressed before I can recommend the manuscript for publication.

Author response: We thank the reviewer for their detailed comments and support. We have addressed the remaining comments below.

(1). Proof sketch of Theorem 2: A brief proof sketch of Theorem 2 should be added to the Methods section of the main text. The authors have used a distinct approach to establish this result, and I believe it would greatly benefit readers to include a concise overview of the underlying reasoning.

Author response: We have added a brief sketch of the proof in Section A of the methods. We also sketch the original proof of classical hardness in this new section.

(2). Figure 1 revision: Figure 1 should be revised to reflect the setting of Theorem 2. Specifically, incorporating elements related to momentum/etc should suffice.

Author response: Thank you for the suggestion. In Figure 1(b), we have added that correlational statistical queries also give rise to classical lower bounds.

(3). Discussion of related literature: I strongly recommend that the authors broaden their discussion of the quantum machine learning literature to include other aspects and clarify how their work connects to prior studies. In particular, the following works should be addressed:

(3.1). Regarding Molteni et al., arXiv:2405.02027 (2024), I agree with the authors' response that the settings differ. In addition, I think the underlying hardness assumptions are distinct. Nevertheless, it would be helpful for readers if the authors explicitly highlight the distinctions between the two works.

Author response: Thank you for your comment. We have added a short discussion of this following the central question, where we emphasize the difference stated in the previous response letter.

(3.2). *It would also be valuable for the authors to discuss how their results relate to the fundamental perspective of quantum barren plateaus, for example as in [Larocca et al., Nature Reviews Physics, 1–16 (2025)] for the general review and [Nietner, arXiv:2310.17716 (2023)] for specific connection to SQ-learning.*

Author response: Thank you for your comment. We clarify that our source of classical hardness is unrelated to barren plateaus in variational quantum algorithms. Barren plateaus are properties of the predictor (e.g., choosing different architectures can mitigate barren plateaus) and can also appear depending on random initialization of parameters. However, our classical hardness is independent of both the predictor and the initialization. Instead, it is a fundamental property of the concept class we are trying to learn. We have decided to not discuss barren plateaus in the present manuscript, as they truly are distinct concepts, and it would likely confuse readers further.

(4). *In response to the authors' answer to my point (11) in the previous round, I think the authors could be more precise about how alternative approaches (i.e., methods other than gradient-based optimization) can be used to solve for the coefficients in Eq. (V3). In practice we only have access to a finite set of training data, so we work with an empirical version of Eq. (V3). In that setting, could one simply apply a linear-regression approach (e.g., Ridge/LASSO) to estimate these coefficients when $N > D$?*

Author response: We apologize if our previous response was confusing. However, we again clarify that our access model does not give us a set of training data directly, so the suggested linear regression approach does not make sense in our setting. An “empirical version” of Eq. (V3) in our access model would mean that we only have access to approximate gradients of the loss function instead of being able to evaluate the loss function exactly, but we nonetheless cannot access the explicit data points directly.

Reviewer #2 (Remarks to the Author):

Thank you for the detailed responses. I am happy with the edits made.

Author response: We thank the reviewer for their positive comments and support.